# Caring for Patients in Need of Palliative Care: Is This a Mission for Acute Care Hospitals? Key Questions for Healthcare Professionals

**DOI:** 10.3390/healthcare10030486

**Published:** 2022-03-06

**Authors:** Paolo Cotogni, Anna De Luca

**Affiliations:** Pain Management and Palliative Care, Department of Anesthesia, Intensive Care and Emergency, Molinette Hospital and University of Turin, C.so Bramante 88/90, 10126 Turin, Italy; adeluca@cittadellasalute.to.it

**Keywords:** palliative care, end-of-life care, emergency department, palliative care team, palliative care unit, aging, healthcare delivery

## Abstract

The prevalence of patients affected by end-stage diseases or advanced cancer is increasing due to an aging population and progression in medicine and public healthcare. The burden of symptoms these people suffer in the last months of life often forces them to seek aid in an emergency department. In developed countries, acute care hospital-based services are often better designed to treat acute clinical conditions than to manage the needs of patients with serious chronic diseases. Thus, the palliative care (PC) population poses very real clinical challenges to healthcare professionals who care for them in hospital settings. The authors have formulated four key questions (who, why, when, and how) to address in order to identify a model for providing the best care for these PC patients. The questions are related to: (1) defining people living with serious chronic diseases; (2) managing the challenge of unplanned hospital admission of these people; (3) identifying PC patients among people with serious chronic diseases; and (4) determining the appropriate work of caring for this inpatient PC population. Clinicians need the knowledge, tools, and services to care for these PC patients, and acute care hospitals should plan the work of caring for these inpatients.

## 1. Introduction

According to the estimates of the World Health Organization, the aging of populations is rapidly accelerating worldwide. By 2050, most people can expect to live into their 60s and beyond. Thus, more and more people will die from chronic rather than acute diseases [1]. The consequences for healthcare systems, their human resources and budgets are profound [2]. In many developed countries around the world, acute care hospital-based services (emergency department, hospital ward, and intensive care unit) are used with surprising frequency by these people during the last year of life [3,4,5]. In particular, the emergency department (ED) has evolved beyond its ‘core business’ to serve as the gateway to hospital resources for palliative care (PC) patients. Several studies suggested that a great part of unplanned visits in the ED made by PC patients were potentially avoidable. In a study, it was found more than half of ED presentations made by patients known to specialist PC services were potentially avoidable [6]. In particular, many of these ED admissions were due to uncontrolled symptoms such as dyspnea, pain, constipation, nausea, and vomiting [6,7]. This can lead to costly hospital services being used to meet chronic care needs and a failure to fully foster the preferences of these individuals. Therefore, this issue poses important concerns and should force all healthcare systems to plan defined actions for caring for these people.

## 2. Methods

Since there are no defined guidelines or recommendations for PC patients admitted to acute care hospitals, the aim of this review is to outline the management of patients with end-stage diseases or advanced cancer admitted to the hospital after an ED visit.

### Search Strategy

A literature search for published manuscripts regarding care of palliative care patients and acute care hospitals across MEDLINE (via PubMed), EMBASE, Cochrane Central, Web of Science, and Scopus was performed. Search was performed using MESH terms and free text words and cross-searching of the two following categories: (1) palliative care patients (“serious chronic diseases” OR “end-of-life care” OR “end-stage diseases” OR “chronic life-limiting diseases” OR “end-stage organ failures” OR “life-threatening illnesses” OR “chronic ill patients” OR “people living with a serious illness”); (2) acute care hospitals (“emergency department” OR “acute care hospital-based services” OR “hospital services” OR “healthcare delivery”). Randomized controlled trials, observational studies, guidelines, and reviews published until June 2020 were included in the search strategy. Search results were limited to publications in English; letters to the editor, case reports, abstracts from conferences, and commentaries were excluded. Screening of all reference lists of relevant studies was also performed in order to identify any missing publications. Searches, reference screening, and study selection were performed independently by the two authors.

This review examines the relevant literature; however, it is not a systematic review but rather a narrative synthesis of studies and systematic reviews. In particular, it assesses a clinical enigma for many clinicians: these are the questions we have been asking in terms of how to manage patients affected by end-stage diseases or advanced cancer seeking aid in an ED. Easy and clear answers are not always available, as PC must be individualized for the patient and those with PC needs are not a homogeneous group. The conundrum in terms of the transition from intensive care to PC in these challenging situations arises only after all other treatment options have been explored and exhausted. The authors of this review have formulated four key questions (who, why, when, and how) which should be addressed in considering the model to deliver the best care for PC patients. These questions form the headings for this review and relate to: (1) defining people living with serious chronic diseases; (2) managing the challenge of unplanned hospital admission of these people; (3) identifying PC patients among people with serious chronic diseases; and (4) determining the appropriate work of caring for this inpatient PC population.

## 3. Results

The search of databases identified 3235 citations (Figure 1). After adjusting for duplicates, 1474 citations remained. Of these, 1295 citations were discarded after reviewing titles and abstracts. The full texts of the remaining 179 studies were examined. Finally, 31 studies met the inclusion criteria and were included in this review.

Four questions are important in order to identify a model to deliver the best care for the PC population. Table 1 presents a short summary of these questions with some key words and comments to consider.

### 3.1. Who Are People with Serious Chronic Diseases?

People with serious chronic diseases are patients with advanced or metastatic cancer as well as those affected by non-oncological illnesses, such as end-stage organ failures (chronic heart failure, chronic obstructive pulmonary disease, liver cirrhosis, and renal disease) or neurological illnesses (amyotrophic lateral sclerosis, Parkinson’s disease, multiple sclerosis, Alzheimer’s and other dementias) [8,9].

Why consider, overall, these people as a unique patient population? Because during the last period of life, patients affected by chronic, progressive, and life-threatening illnesses have been shown to share a similar spectrum of symptoms [8,9]. Moreover, this patient population has the same needs, mainly the relief of symptoms as well as similar psychological, social, and spiritual issues, regardless of the diagnosis [9,10].

The increase in longevity is welcome, but the longer life of people with these diseases is commonly associated with prolonged unpleasant symptoms. Over the past years, symptom management and end-of-life (EOL) care have focused mainly on advanced cancer patients, while less attention has been paid to people affected by end-stage organ failures.

#### 3.1.1. Chronic Diseases

In accordance with the literature [11,12,13,14,15], we propose specific clinical indicators as potential criteria for considering each chronic disease as serious (Table 2).

#### 3.1.2. Chronic Disease-Focused Discussion

Chronic heart failure (CHF) is the most common reason for hospitalization in people aged over 65 [16]. Patients with CHF are characterized by severe symptom burden comparable to that of cancer, poor quality of life (QOL), and poor survival. The use of acute hospital-based services is due to frequent episodes of acute decompensation, leading to patients presenting to ED for symptom control [16].

Patients with chronic obstructive pulmonary disease (COPD) have been shown to have similar scores of severe pain and depression but worse dyspnea, functional status, and anxiety compared with lung cancer patients [17]. However, some studies reported that patients with COPD compared with lung cancer patients received care that was more oriented with prolongation of life than palliation of symptoms during the last 6 months of life, as well as less use of outpatient opiates and benzodiazepines, more frequent admissions to, and greater time spent in, an intensive care unit (ICU) [18].

People with end-stage renal disease (ESRD) have the highest risk for hospitalization among those with chronic diseases [19]. An analysis of a total of 769228 adult patients with ESRD showed that 70% had at least one ED visit during the study period (2005–2011) [20]. The study showed that factors associated with higher rates of ED use included female sex, younger age, black (vs. white) race, comorbidities, institutionalization, Medicaid insurance (vs. Medicare alone), catheter or graft hemodialysis access (vs. fistula), tobacco use, and more recent ESRD diagnosis. The authors also found that the three most common admission diagnoses were hemodialysis access complication, sepsis, and congestive heart failure during the first ESRD year.

Complications occurring during the course of liver cirrhosis can be life-threatening. In patients with decompensated cirrhosis, these frequently are: ascites, gastrointestinal bleeding, hepatorenal syndrome, hepatic encephalopathy, and sepsis [21]. These complications need rapid management, and the ED offers the first opportunity to make a rapid diagnosis and start adequate treatment.

Several studies documented that people with Parkinson’s disease have a high falls injury rate, causing multiple ED presentations, longer ED length of stay, and high admission rate [22].

A secondary analysis of data from a cross-national sample of elderly home care people (≥60 years) was carried out to identify the baseline independent variables predicting risk of ED and hospital use [19]. The authors found that the risk increases as the individual accumulates a list of specific disease diagnoses, clinical problems, and complex treatments. This list includes: the presence of serious disease (renal failure, COPD/emphysema, cancer), infections (pneumonia, urinary tract infection), skin problems (stasis ulcers, wound care), recent deterioration (unintended weight loss, a major deterioration in status in the last 90 days, unscheduled physician visits, falls), and close monitoring for complex treatments (daily nurse monitoring, IV infusion, intravenous medications).

### 3.2. Why People with Serious Chronic Diseases Were Admitted in Acute Care Hospitals?

During the last months of life, most people with chronic diseases experience intense or worsening symptoms, which compel them to seek care in acute care hospitals. These patients are visited in the ED and undergo urgent care for the control of symptoms [7]. Sometimes, patients are brought into the ED because of the distress of family members near the EOL [23].

In developed countries, the proportion of patients with advanced cancer or other chronic diseases who visit the ED during the last months of life is increasing over time, ranging from 40% to 80% [3,7,23]. Barbera et al. observed that pain (mainly abdominal), dyspnea, pneumonia, fatigue, and pleural effusion were the most common reasons for ED visits during both the final 6 months and 2 weeks of life by patients dying of cancer [7]. The authors also found that lung cancer was the most frequent primary cancer diagnosis as well as failure to cope was very common.

In a recent paper, a population-based cohort study was carried out to identify factors influencing ED visits by all adults who died from cancer in England in one year [24]. The authors found that those with greater comorbidity, younger patients, men, Asian and black ethnic minority groups, and people of lower socio-economic status were more likely to visit the ED multiple times in the last month of life. They also reported a relationship between cancer type and multiple ED visits; specifically, patients with a diagnosis of head and neck cancer are at risk of complications that can compromise their airways. Furthermore, those with lung cancer have increased odds of repeated ED visits due to breathlessness being a particularly difficult symptom to manage for patients and their families.

Traditionally, pain is the most studied and publicized symptom experienced by people with serious chronic illnesses; however, many prevalent studies suggest that pain is only one of several distressing symptoms. The others that are frequently prevalent in these patients are: depression, anxiety, confusion/delirium, fatigue, breathlessness/dyspnea, insomnia/sleep disorders, nausea/vomiting, constipation, diarrhea, and appetite loss/anorexia.

Solano et al. analyzed the prevalence of symptoms among end-stage patients suffering from cancer, heart disease, COPD, renal disease, and acquired immunodeficiency syndrome (AIDS) [10]. They reported that three symptoms are particularly common and frequent (i.e., pain, fatigue, and breathlessness), with prevalence often well above 50% in all the investigated diseases. However, insomnia and anorexia are also recurrent symptoms in these patients.

Moens et al. reported that pain, fatigue, anorexia, dyspnea, and anxiety were highly prevalent problems across advanced cancer and non-oncological diseases (heart failure, COPD, renal disease, multiple sclerosis, motor neuron disease, Parkinson’s disease, dementia, and AIDS) [25]. Moreover, the authors describe and compare the prevalence of PC-related problems among these patients showing that there are commonalities in problem prevalence across them.

Patient-reported symptoms are frequently multidimensional in nature and can negatively impact patients’ QOL and performance status as well as increase caregivers’ burden. Knowledge of symptom presence and intensity is important for clinical practice. Therefore, routine comprehensive symptom assessment with the use of validated instruments is strongly recommended in the management of these end-stage diseases. Moreover, symptom management guided by patient-reported instead of clinician-based evaluations is one of the most qualifying aspects of PC [26].

The Edmonton Symptom Assessment System (ESAS), initially developed by Bruera et al. [27], represents one of the first symptom scores in PC; it has since been validated by many studies, translated into over 20 languages, and adopted to manage symptom assessment in several centers worldwide [28]. The use of validated tools, such as the ESAS, can improve the identification of distressing symptoms and lead to better control of these symptoms [29].

In summary, there seems to be a common pathway that patients affected by end-stage chronic diseases have to face. This suggests that PC is relevant for people with all these clinical conditions, although some aspects of management may need modification because of the varying trajectories of functional decline and dependency in non-oncological diseases [30].

### 3.3. When People with Serious Chronic Diseases Should Be Considered Palliative Care Patients?

In the ED, the scope of healthcare providers (HCPs) is the fast identification of signs and symptoms and stabilization of patient’ conditions by rapid interventions [31]. As a matter of fact, the ED has evolved beyond this scope to become ‘the entrance point’ for chronically seriously ill patients who were admitted to the hospital in case of distressing symptoms. Similar to this, the ED becomes a place of care delivery for patients with serious chronic diseases (and their families) in need of a response to physical, psychological, social, and spiritual distress. Included among these distressed patients is the PC population [32].

The crucial question is: how to identify among these patients those in need of PC? The answer is not simple because, in the medical literature, there is no agreement about the definition of the PC population [33]. Moreover, defining when a slowly progressive disease reaches the advanced stages is not always certainly predictable [28]. Actually, clinical observations demonstrate different patterns of functional decline at the EOL among populations with serious chronic diseases [34].

The discussion of selection criteria to identify PC patients is still a controversial topic. Worldwide, this problem exists, and clinicians need practical tools to adequately deal with the PC population in daily practice. In addition, it is not easy for ED physicians to define a patient—whom they are examining for the first time—as a PC patient.

In 2011, Weissmann and Meier reported the conclusions of a consensus panel from the Center to Advance Palliative Care (CAPC), which worked to select criteria by which patients at high risk for unmet PC needs can be identified in advance [11]. The panel developed two checklists of criteria to use for screening, one at the time of admission in the ED and one during the hospital stay as a tool for daily rounds. These criteria were designed to trigger a basic PC screening by the treating staff. The starting point for all criteria chosen by the panelists was the identification of patients with a potentially life-limiting condition.

In fact, the prognosis is one of the critical issues regarding the definition of the PC patient. According to the definition of EOL care by the General Medical Council (UK), ‘’People are ‘approaching the end of life’ when they are likely to die within the next 12 months. This definition includes people with advanced, progressive, incurable conditions or general frailty and co-existing conditions that mean they are expected to die within 12 months” [12].

However, predicting people’s needs is more important than the exact prognostication of their death. The focus is on anticipating patients’ likely needs so that the right care can be provided at the right time. This decision-making leads to better proactive care, in line with the patient’s preferences [12].

The symptom intensity may also be used to trigger the decision-making, such as referral to the PC team for patients with serious chronic diseases. The ESAS has been proposed as a tool for such purpose, although the threshold may need to be refined at each institution [28].

In 2013, the Italian Society of Anesthesia, Analgesia, Resuscitation, and Intensive Care (SIAARTI) released a position paper based on opinions of a panel of experts reporting criteria to identify patients with end-stage chronic organ failures [13]. This paper aims to guide physicians’ decision-making in selecting the best care option—specifically, intensive care or PC for these patients.

Glare et al. proposed a simple 11-item screening tool derived from the National Comprehensive Cancer Network (NCCN) PC guidelines to identify cancer inpatients with greater PC needs who would benefit from referral to a PC team [14]. This screening tool has shown to be a valid tool for screening for PC needs in cancer patients but has the limitation of being disease-specific.

Recently, we have proposed the Simplified Screening Tool (SST) as an easy-to-use, non-disease-specific, and practical instrument to identify patients in the ED for referral to the PC team (Figure 2). This tool could help physicians’ decision-making to appropriately match the healthcare services used with the patients’ needs in the clinical practice [15].

Of note, a review associated a delay or lack in identifying patients in need of PC with a negative impact on care delivery [35]. Indeed, early identification of patients in need of PC by their treating physicians could greatly improve their referral rate to outpatient PC services. Similarly, a better knowledge of the dimension of PC needs in acute care hospitals is crucial to more appropriately matching services to patients’ needs and defining priorities of care [36].

### 3.4. How Acute Care Hospital Should Plan the Work of Caring for This Palliative Care Population?

People with serious chronic illnesses are more likely to use acute hospital-based services towards the EOL (i.e., last weeks or months of life). Obviously, unexpected urgent medical problems lead to unavoidable ED and hospital admissions. However, numbers of ED visits or admission to an ICU near the EOL have been used as indicators of poor-quality care for people affected by serious chronic illnesses, especially for advanced cancer patients [16,17,18]. Thus, understanding why this PC population searches for aid in an ED—and what the extent of the phenomenon is—is critical for determining how best to attempt to minimize the number of patients who visit the ED and, consequently, unplanned hospital admissions [37,38].

In 2006, the American Board of Emergency Medicine had cosponsored hospice and palliative medicine as a recognized medical subspecialty. In 2008, Smith et al., using focus group methodology, evaluated the attitudes, experiences, and beliefs of emergency care providers (i.e., physicians, residents, attending physicians, nurses, social workers, and technicians) about PC in the ED [23]. Specifically, six issues emerged: (1) ED providers equated PC with EOL care; (2) ED providers disagreed about the feasibility and desirability of providing PC in the ED; (3) patients for whom a PC approach has been established often visit the ED because family members are distressed by EOL symptom burden; (4) lack of communication between outpatient and ED providers leads to undesirable decisions (e.g., aggressive and life-sustaining treatments); (5) conflict around withholding life-prolonging treatment is common; and (6) training in pain management is insufficient.

Moreover, the authors described the following potential obstacles to PC in the ED: (1) PC not a major focus of ED providers; (2) emotionally challenging for providers; (3) not being able to ‘act’ is frustrating; (4) environment not appropriate; (5) ED providers do not know patients as well as outpatient providers; (6) patients with PC needs and families sometimes considered a lower priority; and (7) long ED wait times particularly burdensome for patients with PC needs. In summary, ED providers expressed significant discomfort, conflict, and gaps in knowledge with issues concerning PC in the ED.

In 2011, the National Audit Office in the UK estimated that about 25% of all hospital beds are occupied by someone who is dying, and at least 40% of those people have no medical need to be there [12]. In our opinion, there are two key elements that can improve the work of caring for this PC population. Firstly, it is extremely important to identify early (whether in the ED or within the first 24 hours after the hospital admission) the patients in need of PC among those with serious chronic diseases. In our study, we found that more than one-third of patients affected by these diseases awaiting to be hospitalized after an ED visit should be identified as in need of PC [15]. Secondly, the great challenge for healthcare professionals is to be prepared to manage the transition from curative care (‘active treatment’) to PC appropriately.

In the last decade, the interest and knowledge of emergency care providers about PC are markedly increased. Similarly, there has been an explosion of clinical studies comparing concurrent curative care and PC versus usual care alone in outpatient and inpatient settings, providing good evidence to support the integration of PC along the disease trajectories of serious chronic illnesses [9,39,40,41,42].

Gott et al. carried out the first study exploring how transitions to a PC approach were perceived to be managed in acute hospital settings in the UK [43]. The authors found that a structured approach to transitions to PC was far from the reality of practice in these settings. Based on these data, the UK policy guidance recommended in acute hospitals to improve early identification of people in the last year of life, helping to reduce hospitalizations and facilitate accessing supportive and PC services.

New models of care that better manage ED presentations of patients bearing serious chronic diseases were proposed in order to potentially reduce avoidable hospitalizations and ED representations [44]. One model provides that basic palliative skills should be known by ED professionals and readily applied on a daily basis to effectively treat these patients, even in a busy ED [31,45,46]. Similarly, the provision of early and simultaneous PC in cancer patients could be enhanced by increasing the delivery of primary PC by oncologists. In view of the growing numbers of patients who could benefit, the emphasis of the UK strategies is on improving EOL care delivered by primary care teams, hospital staff, and social care services [47]. In this way, PC principles and practices are integrated into any healthcare setting and delivered by treating professionals involved in the daily care of patients with serious chronic illnesses.

However, the weakness of this model is that most hospital HCPs do not usually receive adequate training on pain management and PC, as well as on communication, especially about EOL care in dying patients. Indeed, PC and EOL care are not easy conversations to have between the emergency (or treating) physician and the patient (or caregivers). In addition, many physicians relate that they have encountered significant ethical dilemmas in handling refractory symptoms and withdrawing life-sustaining treatments in dying patients [48].

For 20 years, in order to better manage patients with advanced cancer or end-stage diseases, specialist hospital-based PC programs have been developed, specifically the creation of interdisciplinary PC teams [49,50]. These PC teams are composed of physicians, nurses, and psychologists, with the collaboration of other HCPs (dietician, respiratory therapist, physical therapist, and pharmacist) as well as social workers, case managers, spiritual counselors, and volunteers. All the members of the team have the professional qualifications, training, and skills to deliver optimal patient- and family-centered care.

The members of the PC team comprehensively address the multidimensional care needs of patients and their families. In detail, the scope of PC teams is to improve symptoms management, increase awareness of prognosis and treatment options, explain goals of EOL care, and assist in the planning of discharge [51]. Over the past decade, PC programs have grown by more than 150% in the USA, such that in 2014 almost 90% of hospitals with 300 beds have a PC team [9].

Multiple studies found that an ED-initiated PC consultation can determine benefits on outcomes in serious chronic patients, e.g., significantly shorter length of stay [52], improved QOL without shortening survival [53], and timely referral to community, inpatient PC teams, or hospice services [54,55].

Several randomized controlled trials and case-control studies compared the outcomes in patients who were referred to hospital-based PC teams with the outcomes in patients who received usual/standard care, in people with advanced cancer [56], neurologic disease, lung disease, and older individuals with multiple co-existing conditions and frailty [9]. Altogether, these clinical investigations found decreases in symptoms burden, spiritual distress, healthcare services used, and costs, as well as improvements in patients’ and families’ QOL and satisfaction among the people that were referred to PC services [57,58,59,60,61,62].

Over the last decade, a new model of care in managing the PC population (mainly advanced cancer patients) [63] in the hospital setting has been represented by the acute palliative care unit (APCU) [64]. The purpose of the APCU is to obtain rapid and effective control of high scores physical symptoms and provide intensive psychosocial support in case of severe distress. The APCU is characterized by a shorter length of stay (i.e., 10–15 days) and a lower death rate (i.e., 40–50%) compared to in hospice ones [65]. Thus, the APCU differs from the hospice, which offers reduced medical interventions, more extensive long-term stay (i.e., 1–6 months), and normally near-death care, with a mortality rate near 95% [66].

In our tertiary care hospital, the PC team must have previously assessed all patients to be admitted to the APCU. Priority for admission is based on the acuity of need, which is determined by the members of the team using a clinical screening scale for symptoms and distress. The physician—together with the nurse—discusses the admission to the APCU with the patient and the family as well as with the treating physician.

The APCU delivers multidisciplinary PC for patients and their caregivers, with the goal of improving QOL, managing transition to EOL care [67], and arranging appropriate hospital discharge. The discharge is a crucial choice, as it has to provide a place (e.g., home, hospice, or long-term care facility) where the person can obtain proper care until death, avoiding ED (re)presentations for insufficient symptoms’ management [35]. In our healthcare system, the choice of the discharge location depends mainly on the patient’s preferences but also on the availability of caregivers, life expectancy, and performance status.

To summarize, the APCU is an acute hospital-based service that offers the PC population comprehensive and ‘personalized’ care tailored to needs and preferences.

## 4. Limitations of the Study

The present study has several limitations. First, our study is not a systematic review but rather a narrative review of the included studies. Second, our study focused on addressing four questions in order to identify a model for delivering the best care to the PC population. However, other effective interventions or integrated strategies might be important to improve care for PC patients presenting to acute care hospitals. In addition, the questions could be drawn using a different methodology. Third, we did not choose a disease-focused discussion of the four questions but rather have considered these individuals overall as a unique patient population. Indeed, it was shown during the EOL that people living with serious chronic illness share a similar spectrum of symptoms, as well as have the same needs. Finally, clinicians need practical tools and pathways to provide EOL care to acute PC patients outside the ICU. For this purpose, our review proposes the SST as a practical tool to identify patients in the ED for referral to the PC team, as well as a dedicated hospital unit for acute PC patients. However, suggesting a disease-centered or a practical, comprehensive approach to EOL care is beyond the aims of this paper.

## 5. Future Directions for Palliative Care Research

Limited evidence exists pertaining to research priorities across PC [68]. However, some descriptive themes are described briefly below.

### 5.1. Research

Data over the past decade show an increase in the number of PC researchers and publications, as well as a large body of evidence demonstrating the benefits of early PC in many serious chronic diseases. However, we need to call for more large-scale, prospective studies of the utilization and outcomes associated with PC. Moreover, future attempts to identify a model for providing the best care for PC patients should involve multidisciplinary representation of healthcare providers and other stakeholders, as well as PC patients, families, and caregivers; such inclusion will provide reliability and improve the feasibility of the developed priorities.

### 5.2. Interventions

Key interventions to create a new model of care for patients with serious chronic diseases in acute care hospitals include: the use of validated screening tools to identify people with unmeet PC needs and organizational decision-making to manage unplanned hospital admissions.

In addition, there is a strong and urgent call for interventions to promote equitable access to quality palliative and EOL care tailored to meet the patient’s needs, especially for non-cancer patients.

Finally, there is a need to improve all aspects of communication in PC. This issue includes communication among clinicians, patients, and families/caregivers as well as among clinicians of different specialties and services (i.e., acute care hospital and community-based PC services). Moreover, in hospitals without an APCU, it should be possible, through a consultation process, to transfer acute PC patients to a hospital with an available APCU.

### 5.3. Education

Training of non-PC clinicians is a critical area for research, with primary care and ED staff as the main group of HCPs who should be targeted for additional education in PC. In particular, continued medical education, as well as up-to-date resources on pain and symptom management in EOL care, may enhance PC delivery in everyday practice. In addition, clinician education on how and when to best communicate both the transition from active treatment to PC and the patient’s prognosis should be improved.

## 6. Conclusions

The prevalence of people affected by serious chronic diseases is growing due to population aging and progresses in medicine and public healthcare. This change occurring in developed countries determines important organizational challenges due to increasing demand for acute hospital-based services (especially ED). Thus, it is crucial to evaluate the present characteristics and trends of acute care hospital use by these people for efficient management of limited healthcare resources and allocation of HCPs.

Every acute care hospital should rethink the actual work of caring for inpatients with serious chronic diseases managing the transition from curative treatments to PC. The PC should be made available based on the clinical problems (mainly, presence and intensity of symptoms) and needs (mainly, psychological, social, and spiritual support) patients have, and not based on their diagnosis or prognosis.

The model for the best care of patients with end-stage diseases should be based firstly on the early identification of the PC population in all hospital settings. Within acute care hospitals, the primary model of PC delivery is the interdisciplinary PC team. In addition, an important model of care in managing these patients is represented by the APCU. In this review, we discussed four key questions for HCPs that summarize a stepwise algorithmic approach to PC delivery (Figure 3). In conclusion, expanding, facilitating, and anticipating access to PC services is the right pathway to meet the needs of the PC population. Indeed, well-timed PC is potentially ‘preventive care’ to lessen patients’ and families’ crises at the EOL, which are the main reasons for avoidable ED visits and hospital admissions.

## Figures and Tables

**Figure 1 healthcare-10-00486-f001:**
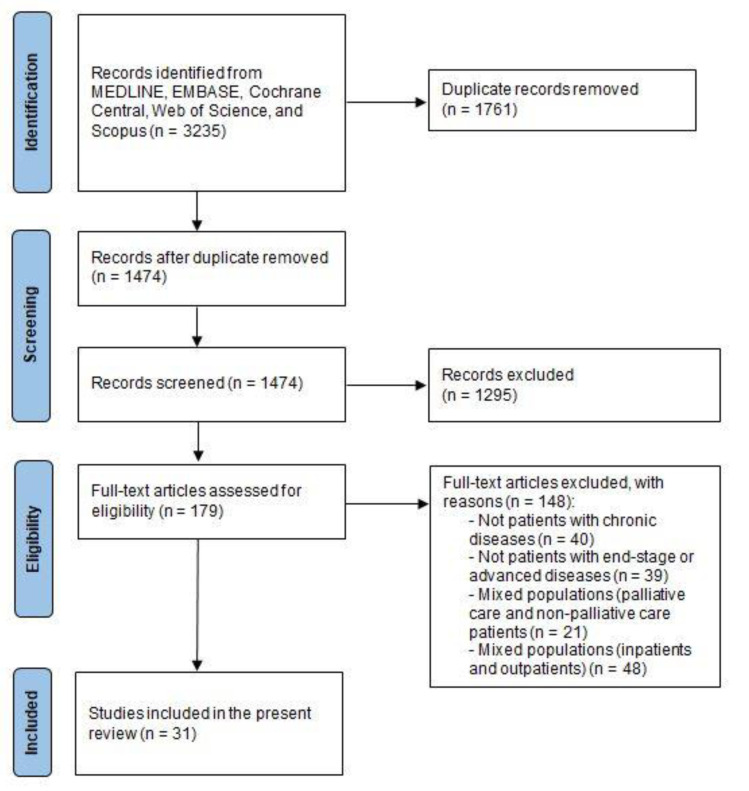
PRISMA flow diagram.

**Figure 2 healthcare-10-00486-f002:**
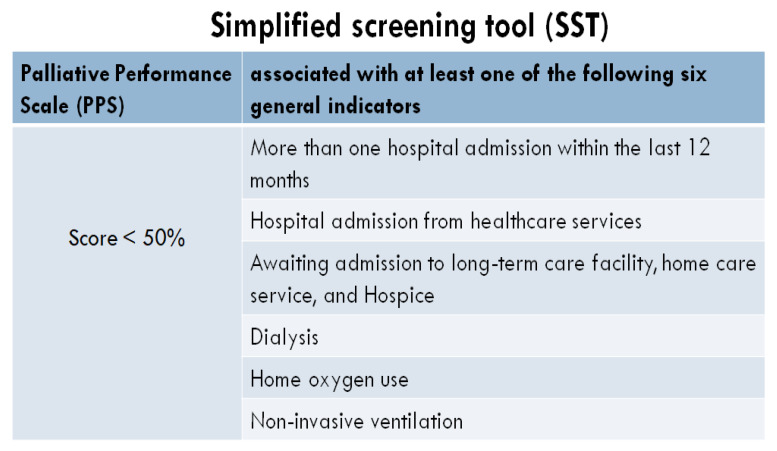
Simplified Screening Tool.

**Figure 3 healthcare-10-00486-f003:**
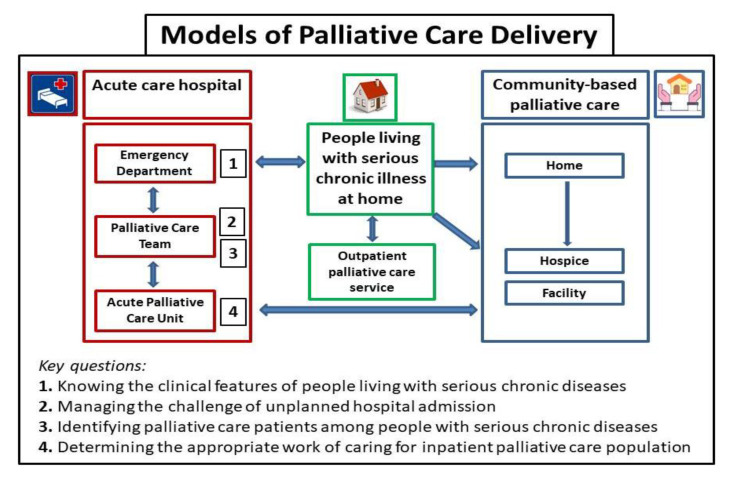
Models of palliative care delivery.

**Table 1 healthcare-10-00486-t001:** A brief summary of the questions to be addressed to identify a model for delivering the best care to the palliative care population.

Keyword	Question	Comments
Chronic diseases	**Who** are people with serious chronic diseases?	Assess presence and intensity of distressing symptoms and their impact on patient QOL
Unplanned hospital admission	**Why** people with serious chronic diseases were admitted in acute care hospitals?	Assess distress both of patient and family members Consider socio-economic status
Palliative care patient	**When** people with serious chronic diseases should be considered PC patients?	Consider both patients with advanced cancer and non-cancer diseases Screen patients for unmeet PC needs Evaluate performance status, trajectories of functional decline, number of ED visits and hospitalizations, prognosis
Hospital care	**How** acute care hospital should plan the work of caring for this PC population?	Introducing the use of the bio- psychosocial model approach Supporting HCP training in PC and communication Developing the hospital-based PC team and acute palliative care unit Considering home or hospice care according to patient and family preferences

ED, emergency department; HCP, healthcare professional; PC, palliative care; QOL, quality of life.

**Table 2 healthcare-10-00486-t002:** Clinical indicators specific for each chronic disease.

**Heart failure**	NYHA class IV ≥1 admission within the last 6 months Hypotension and/or fluid retention Dependency from intravenous medications Poor response to cardiac resynchronization therapy Cachexia	**Dementia**	Unable to walk without assistance, urinary and fecal incontinence, no consistently meaningful conversation Considerable assistance required Urinary tract infection, pressures sores (stage 3–4), recurrent fever Reduced oral intake, cachexia, aspiration pneumonia
**Respiratory failure**	>70 years Forced Expiratory Volume in the first sec <30% predicted Dyspnea scale grade of ≥3/Dependency from oxygen use ≥1 admission within the last 12 months Cachexia Considerable assistance required	**Stroke**	NIH Stroke Score ≥20 for left and ≥15 for right stroke Onset of headache + nausea/vomiting within 6 hCT scan showing a middle cerebral artery stroke ≥50% >75 years, history of ictus, fever, atrial fibrillation Conjugate deviation of the eyes Early reduced level of consciousness
**Liver failure**	Reduced oral intake, cachexia, aspiration pneumonia Deemed ineligible for transplantation Model for end-stage liver disease (MELD) score >25 Refractory ascites, spontaneous peritonitis, recurrent variceal bleeding, hepatorenal syndrome, hepatic encephalopathy	**Parkinson’s disease**	Decreasing response to treatments/medications Considerable assistance required Less controlled disease (increasing ‘off’ periods) Dyskinesias, mobility problems, falls Dysphagia (moderate/severe), psychiatric signs
**Renal failure**	>75 years Advanced cancer, heart or lung failure, vegetative state, dementia, cachexia	**Amyotrophic lateral sclerosis**	Patient/family requests for information/help about disease/symptoms Pain requiring high dosages of analgesics Need for a feeding tube Dyspnea or hypoventilation with vital capacity <50% Loss of function in at least two body regions Communication difficulties Weakness Cognitive difficulties Recurrent infections Dysphagia, cachexia, aspiration pneumonia
**Cancer**	Uncontrolled symptoms Moderate-to-severe cancer-related distress Metastatic solid tumors, central nervous system metastasis ECOG ≥3 or Karnofsky Performance Status <50 Persistent hypercalcemia, delirium, superior vena cava syndrome, spinal cord compression Cachexia Malignant effusions Palliative stenting or venting gastrostomy Patient/family concerns about disease/decision-making Patient/family requests for palliative care		

ECOG, Eastern Cooperative Oncology Group; NIH, National Institutes of Health; NYHA, New York Heart Association.

## Data Availability

Not applicable.

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
