# Peer review of "Caring for Patients in Need of Palliative Care: Is This a Mission for Acute Care Hospitals? Key Questions for Healthcare Professionals"

_healthcare, 2022, doi:10.3390/healthcare10030486_

Round 1

Reviewer 1 Report

I find this narrative review very interesting and a recent important topic has been discussed. However, I have not found criteria for ethical and practical selection of palliative care patients in the clinical practice. I would choose   disease focused discussion (cardiac, COPD, cancer and neurological) and I would discuss the four questions on this way.

Another important part could be the discussion of the selection criteria of acute palliative care patients. I accept that these problems exist worldwide, but I need practical issues if I would like to deal with the palliative care adequately.  As a clinician I need practical techniques, otherwise I will offer these acute care patients to the ICU and they will provide end of life care. This reviewer suggests a disease centered and a practical approach to the end of life care.

Author Response

I find this narrative review very interesting and a recent important topic has been discussed.

R. Thank you for your appreciation.

However, I have not found criteria for ethical and practical selection of palliative care patients in the clinical practice.

R. Many thanks for the appreciated suggestion. According to the literature, we have now proposed specific clinical indicators as potential criteria for considering each chronic disease as serious (Table 2) (pages 4 and 5).

I would choose disease focused discussion (cardiac, COPD, cancer and neurological) and I would discuss the four questions on this way.

R. Thank you for this valuable suggestion. We have now added an additional section (ie, 3.1.2) that reports a “Chronic disease-focused discussion” (page 6).

Nevertheless, we did not choose a disease-focused discussion of the four questions, but rather have considered these individuals overall as a unique patient population. Indeed, it was shown during the EoL that people living with serious chronic illnesses share a similar spectrum of symptoms, as well as have the same needs. However, with respect to the reviewer's comment, we have reported this choice as a limitation of the study (Section 4: Limitations of the Study) (page 13).

Another important part could be the discussion of the selection criteria of acute palliative care patients. I accept that these problems exist worldwide, but I need practical issues if I would like to deal with the palliative care adequately. As a clinician I need practical techniques, otherwise I will offer these acute care patients to the ICU and they will provide end of life care. This reviewer suggests a disease centered and a practical approach to the end of life care.

R. We absolutely agree with the reviewer. We have added the following sentences: “The discussion of selection criteria to identify PC patients is still a controversial topic. Worldwide, this problem exists and clinicians need practical tools to adequately deal with the PC population in daily practice.” (page 9).

In addition, we have now added a figure presenting the Simplified Screening Tool (SST) which is an easy-to-use, non-disease-specific, and practical instrument to identify patients in the ED for referral to the PC team (Figure 2) (page 10).

Finally, we have now added a new section (Section 4: Limitations of the study) (page 13) in which we report the following limitation of the study: “Finally, clinicians need practical tools and pathways to provide EoL care to acute PC patients outside the ICU. For this purpose, our review proposes the SST as a practical tool to identify patients in the ED for referral to the PC team, as well as a dedicated hospital unit for acute PC patients. However, suggesting a disease-centered or a practical comprehensive approach to EoL care is beyond the aims of this paper.”

Reviewer 2 Report

The subjects this article dealt with are very important for health professionals. However, the authors reviewed the relevant literature and answered four questions without appropriate methodology. Furthermore, readers cannot assume how the four questions were drawn. This would be more appropriate as a review article rather than an original article. 

Author Response

The subjects this article dealt with are very important for health professionals.

R. Thanks for your comment.

However, the authors reviewed the relevant literature and answered four questions without appropriate methodology. Furthermore, readers cannot assume how the four questions were drawn.

R. Thank you for this critique. We have added the results of our search strategy (page 2). Also, we have now added a figure presenting the PRISMA flow diagram (Figure 1) (page 3).

In our opinion, these four key questions summarize a stepwise algorithmic approach to the management of these patients. In order to improve the understandability of our approach to palliative care delivery, we have added a graphical abstract summarizing it. (Figure 3) (page 14).

Finally, we have now added a new section (Section 4: Limitations of the study) (page 13) in which we report the following limitations of the study: “First, our study is not a systematic review, but rather a narrative review of the included studies. Second, our study focused on addressing four questions in order to identify a model for delivering the best care to the PC population. However, other effective interventions or integrated strategies might be important to improve care for PC patients presenting to acute care hospitals. In addition, the questions could be drawn using a different methodology.”

This would be more appropriate as a review article rather than an original article.

R. Thank you for this suggestion. We have revised and edited our manuscript as a review instead of an article.

Reviewer 3 Report

In this review authors raise a very interesting a challenging question about the management of patients with end staged diseases. They present a wide spectrum of the current literature along with their points of view. It is an interesting work, but to my opinion there are several points that can be improved.

INTRODUCTION

Authors should add numbers about the frequency of unplanned visits by PC patients in ED. Moreover, they should also provide the main reasons (symptoms) that lead these patients to the ED.

METHODS

I cannot understand the reason behind the literature search.  Authors present a search strategy, but its results are not presented subsequently. They should either omit the literature search strategy or present its results.

RESULTS

All subsections are numbered as 3.1

3.1 SECTION- WHO

More than half of this section reports symptoms associated with serious chronic diseases. According to the manuscript’s structure this seems to be mostly the topic of the 3.2 section. In this section I believe that authors should provide more information about which diseases are considered as chronic. Furthermore, they should emphasize on the turning points that render each disease as an end stage disease.

3.3 SECTION- PALLIATIVE CARE CONSIDERATION

Unfortunately, there is no consensus about the definition of palliative care patients.  Nevertheless, I don’t think that ED physicians are responsible to judge a patient -that they probably examine for first time- as a PC patient.

Authors should emphasize more on the need to identify PC patients in advance by their treating physicians.  

3.4 SECTION- ACUTE CARE OF PC PATIENTS

Authors present well the difficulties of ED physicians to face PC patients and their families. Indeed, APCU seems a reasonable option which could optimize the management of PC patients.

But, what do the authors and the literature propose for hospitals without APCUs? Is it possible to create a network (probably 24/7) to transfer these patients to hospitals with available APCUs??

CONCLUSION

In the results authors presented that currently there are several unmet need regarding the identification and the management of PC patients. They should include in this section future directions for researchers.   

To my opinion these four questions summarize a stepwise algorithm approach for the management of these patients. It would enhance understandability to provide a corresponding figure, probably as graphical abstract.

Author Response

In this review authors raise a very interesting a challenging question about the management of patients with end staged diseases. They present a wide spectrum of the current literature along with their points of view. It is an interesting work, but to my opinion there are several points that can be improved.

R. Thank you for your appreciation.

INTRODUCTION

Authors should add numbers about the frequency of unplanned visits by PC patients in ED. Moreover, they should also provide the main reasons (symptoms) that lead these patients to the ED.

R. Many thanks for the suggestion. We have added the information about the frequency of unplanned visits by PC patients in ED and the main reasons (symptoms) that lead these patients to the ED (page 1).

METHODS

I cannot understand the reason behind the literature search. Authors present a search strategy, but its results are not presented subsequently. They should either omit the literature search strategy or present its results.

R. Many thanks for this valuable suggestion. We have added the results of our search strategy (page 2). Also, we have now added a figure presenting the PRISMA flow diagram (Figure 1) (page 3).

RESULTS

All subsections are numbered as 3.1

R. Thank you. We have corrected the typo.

3.1 SECTION- WHO

More than half of this section reports symptoms associated with serious chronic diseases. According to the manuscript’s structure this seems to be mostly the topic of the 3.2 section. In this section I believe that authors should provide more information about which diseases are considered as chronic.

R. Thanks for your careful review. According to your suggestion, we have moved the information about symptoms associated with serious chronic diseases from section 3.1 to section 3.2 (pages 7 and 8), as well as added an additional section (ie, 3.1.2) that reports a “Chronic disease-focused discussion” (page 6).

Furthermore, they should emphasize on the turning points that render each disease as an end stage disease.

R. Many thanks for the appreciated suggestion. According to the literature, we have now proposed specific clinical indicators as potential criteria for considering each chronic disease as serious (Table 2) (pages 4 and 5).

3.3 SECTION- PALLIATIVE CARE CONSIDERATION

Unfortunately, there is no consensus about the definition of palliative care patients. Nevertheless, I don’t think that ED physicians are responsible to judge a patient -that they probably examine for first time- as a PC patient.

R. We absolutely agree with the reviewer. We have added the following sentences: “The discussion of selection criteria to identify PC patients is still a controversial topic. Worldwide, this problem exists and clinicians need practical tools to adequately deal with the PC population in daily practice. In addition, it is not easy for ED physicians to define a patient -who they are examining for the first time- as a PC patient.” (page 9).

Authors should emphasize more on the need to identify PC patients in advance by their treating physicians.

R. Thank you for the suggestion. We have added the following sentence: “Indeed, early identification of patients in need of PC by their treating physicians could greatly improve their referral rate to outpatient PC services.” (page 10).

3.4 SECTION- ACUTE CARE OF PC PATIENTS

Authors present well the difficulties of ED physicians to face PC patients and their families. Indeed, APCU seems a reasonable option which could optimize the management of PC patients.

But, what do the authors and the literature propose for hospitals without APCUs? Is it possible to create a network (probably 24/7) to transfer these patients to hospitals with available APCUs??

R. We agree with the reviewer. We have added the following sentence: “Moreover, in hospitals without an APCU, it should be possible, through a consultation process, to transfer acute PC patients to a hospital with an available APCU.” (page 13, section 5.2).

CONCLUSION

In the results authors presented that currently there are several unmet need regarding the identification and the management of PC patients. They should include in this section future directions for researchers.

R. Thank you for the suggestion. We have now added a new section (Section 5. Future directions for palliative care research) (pages 13 and 14).

To my opinion these four questions summarize a stepwise algorithm approach for the management of these patients. It would enhance understandability to provide a corresponding figure, probably as graphical abstract.

R. Many thanks for this valuable suggestion. We have now added a graphical abstract that summarizes a stepwise algorithmic approach to PC delivery (Figure 3) (page 14).

Reviewer 4 Report

The manuscript is clear, properly assigned to the field, prepared in accordance with the publishing guidelines. The authors correctly identified the knowledge gap, and this is the most recent and interesting overview for the entire scientific community.
References are valid for most of the last 5 years, you can limit the literature over 5 years to the necessary items.
The manuscript is scientifically substantiated, very legible, well-written methodology. The results are presented correctly and described correctly and are therefore easy to understand.

Author Response

The manuscript is clear, properly assigned to the field, prepared in accordance with the publishing guidelines. The authors correctly identified the knowledge gap, and this is the most recent and interesting overview for the entire scientific community.
References are valid for most of the last 5 years, you can limit the literature over 5 years to the necessary items.
The manuscript is scientifically substantiated, very legible, well-written methodology. The results are presented correctly and described correctly and are therefore easy to understand.

R. Many thanks for your appreciation!

Round 2

Reviewer 3 Report

Thank you for adressing my comments and resubmitting a much improved version of the original manuscript. I hane no more comments.